# Translation of SARS-CoV-2 gRNA Is Extremely Efficient and Competitive despite a High Degree of Secondary Structures and the Presence of an uORF

**DOI:** 10.3390/v14071505

**Published:** 2022-07-08

**Authors:** Lionel Condé, Omran Allatif, Théophile Ohlmann, Sylvain de Breyne

**Affiliations:** CIRI, Centre International de Recherche en Infectiologie, (Team Ohlmann), Univ Lyon, Inserm U1111, Université Claude Bernard Lyon 1, CNRS UMR5308, ENS de Lyon, F-69007 Lyon, France; lionel.conde@ens-lyon.fr (L.C.); omran.allatif@ens-lyon.fr (O.A.)

**Keywords:** translation initiation, SARS-CoV-2, eIF4F, mRNA, uORF

## Abstract

The SARS-CoV-2 infection generates up to nine different sub-genomic mRNAs (sgRNAs), in addition to the genomic RNA (gRNA). The 5′UTR of each viral mRNA shares the first 75 nucleotides (nt.) at their 5′end, called the leader, but differentiates by a variable sequence (0 to 190 nt. long) that follows the leader. As a result, each viral mRNA has its own specific 5′UTR in term of length, RNA structure, uORF and Kozak context; each one of these characteristics could affect mRNA expression. In this study, we have measured and compared translational efficiency of each of the ten viral transcripts. Our data show that most of them are very efficiently translated in all translational systems tested. Surprisingly, the gRNA 5′UTR, which is the longest and the most structured, was also the most efficient to initiate translation. This property is conserved in the 5′UTR of SARS-CoV-1 but not in MERS-CoV strain, mainly due to the regulation imposed by the uORF. Interestingly, the translation initiation mechanism on the SARS-CoV-2 gRNA 5′UTR requires the cap structure and the components of the eIF4F complex but showed no dependence in the presence of the poly(A) tail in vitro. Our data strongly suggest that translation initiation on SARS-CoV-2 mRNAs occurs via an unusual cap-dependent mechanism.

## 1. Introduction

In eukaryotes, translation is a highly regulated step of gene expression in which mRNAs are decoded by ribosomes into proteins. This mechanism can be divided into four steps: initiation, elongation, termination and ribosome recycling. The initiation process is the rate-limiting step that promotes the formation of ribosomal complexes at the initiation codon [1]. The majority of the cellular and viral mRNAs use the scanning model to engage the ribosome on the mRNAs. This starts with the formation of a 43S complex, which is composed of the 40S small ribosomal subunit associated with several eukaryotic initiation factors (eIFs), notably eIF1, eIF1A, eIF3 and the ternary complex (TC) [1,2]. This TC results in the association of eIF2 with a GTP molecule and the Met-tRNA_i_^Met^ [3]. Most eukaryotic mRNAs are capped at their 5′ extremity and polyadenylated at their 3′ end in order to ensure integrity, stability and efficient translation of the mRNA. The cap structure consists of a 7-methylguanylate residue connected to the mRNA via an unusual 5′ to 5′ triphosphate linkage [4]. The cap interacts with the eIF4F complex, which is composed of the cap-binding protein eIF4E, the scaffold protein eIF4G and the RNA helicase eIF4A. The eIF4F complex plays a crucial role during initiation because it establishes a physical and functional link between the mRNA (through a cap–eIF4E interaction) and the 43S complex (through an eIF4G–eIF3 interaction) [5,6,7]. Once bound to the 5′end of the mRNA, the 43S complex scans the 5′ untranslated region (UTR) in a 5′ to 3′ direction to reach the cognate AUG start codon. This is catalyzed by the RNA and ATP-dependent helicase activity of eIF4A, which unfolds the RNA structures encountered within the 5′UTR. Recognition of the translation initiation site (TIS) is ensured by a perfect base pairing between the AUG codon and the anticodon of the tRNA_i_^Met^, which triggers GTP hydrolysis and eIFs release, followed by association of the 40S ribosomal subunit with the 60S ribosomal subunit to form the 80S ribosome competent for elongation [1,2,8].

The efficiency of translation initiation is largely regulated by both eIF2 and eIF4F [9,10,11], but the intrinsic characteristics of the mRNA, such as length, RNA folding and upstream AUG codons, can also modulate the flow and quantity of ribosomes on the transcript [12]. In general, initiation on most eukaryotic mRNAs adheres to the first AUG rule, in which the AUG triplet nearest the 5′end is the beginning of the reading frame [13,14,15,16]. It is noteworthy that the nucleotides that surround the initiation codon are critical for efficient recognition with the following sequence—GCCA/GCCAUGG—being optimal and defined as the Kozak context [13,14]. It has been reported that the nucleotides upstream of the AUG codon, especially the purine at position −3 (according to the position of the AUG codon), are important for efficient initiation [13,17,18,19]. UV cross-linking experiments in the 48S pre-initiation complex (PIC) strongly suggested an interaction between eIF2α and the −3 purine [20]. This was confirmed later by cryo-EM analyses of PIC indicating that the arginine residues (Arg55 and Arg57) of eIF2α are located within close proximity of the mRNA nucleotides upstream to the AUG codon, including the −3 position [21]. These data are consistent with the role of eIF2 in AUG codon recognition [22]. However, the influence of the nucleotide in the +4 position is rather controversial. Some experiments showed that mutation of the guanosine reduces translation initiation efficiency [13], but it was shown by analysis of all possible combinations using a FACS-seq approach that the +4G was not always the optimum [23]. In addition, an alternative explanation was suggested and it was based on the amino acid constraint on the second codon that participates in post-translational modification, such as the N-terminus methionine excision [24,25]. Thus, according to the robustness of this Kozak context, the AUG codon can be more or less recognized as a translation initiation site or even completely ignored. In addition to these features, the presence of short upstream open reading frames (uORFs) within the 5′UTR can also have an impact on translational efficiency of the main gene due to the fact that some ribosomes will translate these uORFs and then dissociate from the mRNA before reaching the main reading frame [26,27,28]. This is why the presence of these uORFs is often associated with a reduction of production of the main protein product. Interestingly, in some particular physiological conditions, the level of active TC is reduced and this favors the bypass of these uORFs, resulting in efficient translation of the main ORF [29,30,31].

Another key feature of the mRNA that can regulate ribosome recruitment is the RNA structure adopted by the 5′UTR. It is now well known that a thermodynamically stable hairpin structure at the 5′end can completely block 43S complex attachment to the cap structure [32,33,34,35,36] and it usually requires additional RNA helicases to overcome the energy barrier imposed by these structures [37,38,39]. In some cases, these RNA motifs can serve as anchors for the binding of specific proteins, creating an RNA-protein complex that can arrest the scanning of the 43S ribosome. The best-known example remains the mRNA coding for ferritin, in which the 5′UTR contains an iron-responsive element, which is a short and conserved stem-loop that is recognized when the endogenous iron concentration becomes low in order to down-regulate translation of ferritin [40,41,42,43]. In RNA positive-strand viruses, there are often many stable RNA structures that are involved in different steps of viral replication (dimerization, packaging, export, etc.) [44,45,46] and can interfere with viral translation. As such, viruses often develop strategies to council replication and translation. The best example comes from picornaviruses, which contain uncapped mRNAs harboring a very long and highly structured 5′UTR with multiple upstream AUG codons that are not used for translation [47,48]. Indeed, the RNA folding adopted by the 5′UTR has the capacity to create a functional structure that directly promotes ribosome recruitment and was defined as internal ribosome entry segment (IRES) [49]. IRESes are not restricted to picornaviruses and are present in other viral families [50,51] and cellular mRNAs [51,52,53]. Viral IRESes remain the most studied and despite differences in length, sequence and structures, they usually have the characteristics to directly interact with some components of the 43S initiation complex to bring the ribosome close to the initiation codon [54,55,56,57,58,59,60,61,62,63]. On the other hand, eukaryotic IRESes concern ~10% of the mammalian mRNAs and they are generally less structured than viral IRESes [51]. Overall, this mode of initiation is cap- and eIF4E-independent and can become the predominant mode of translation initiation mechanism during cellular stresses and viral translational escape [51,64,65,66,67].

The severe acute respiratory syndrome coronavirus 2 (SARS-CoV-2) [68] contains a long and positive RNA genome (gRNA), which is capped and polyadenylated. Once in the cytoplasm, the gRNA is immediately translated into two large polyproteins, ORF1a and ORF1ab, which generate 16 non-structural proteins (nsp) after proteolytic cleavage. Expression of these non-structural proteins is essential for replication and this starts with the transcription of a full-length negative-sense genomic copy that serves as template for the production of positive-sense genomic RNAs. One of the peculiarities of the coronaviruses is that they harbor a conserved leader sequence located at the 5′ end of the transcript that serves as a *cis*-acting element for the transcription of all sub-genomic mRNAs (sgRNA). This leader sequence is added to all sgRNAs through a polymerase template switching that occurs when the viral polymerase encounters the transcriptional regulatory sequence (TRS). The TRS precedes every single viral gene and is called the body TRS (TRS-B), with a conserved core sequence that is thought to hybridize with the leader TRS (TRS-L) [69,70,71]. As a result, all 5′UTRs of SARS-CoV-2 mRNAs (sgRNAs and gRNAs) share an absolutely conserved 75-nucleotide (nt.) stretch, which is called the leader sequence. This leader sequence is then followed by a variable RNA length (from 0 to 190 nt.) that terminates onto an AUG start codon (Figure 1). Viral transcription generates up to nine sgRNAs coding for the structural (S, E, M and N) and accessory (ORF-3a, ORF-6, ORF-7a, ORF-7b and ORF-8) proteins (Figure 1) [71,72]. The 5′UTRs of ORF-7a and ORF-8 gRNA are identical and harbor only the leader sequence (Figure 1 and Table 1). The 5′UTRs of ORF-S, ORF-3a, ORF-E and ORF-N are composed of the leader sequence followed by only a few nucleotides (from 1 to 8 nt.) (Figure 1 and Table 1). In contrast, the 5′UTR of ORF-M, ORF-6 and ORF-7b are relatively long, with respective lengths of 119, 230 and 151 nt. including the leader sequence (Figure 1 and Table 1). Thus, every single sgRNA harbors its own 5′UTR and the nucleotide surrounding context of each AUG start codon is also unique (Table 1); these structural features could potentially modulate translational efficiency.

The 5′UTR of the gRNA is the longest (265 nt.) of all the viral mRNAs and contains five stem-loop structures (SL1 to SL5) that have recently been modelized by chemical probing analyses (Figure 2) [73,74]. These SL are critical for replication, RNA synthesis and escape from nsp1-mediated translation inhibition [75,76,77,78,79,80,81]; however, they could potentially represent hurdles for translation, notably at the level of ribosome recruitment and scanning. Although the viral genome of SARS-CoV-2 is capped and polyadenylated, the translation initiation mechanism used to locate the AUG start codon inside the SL5 remains undetermined. In particular, one can predict that the high level of RNA secondary structures within the 5′UTR could be a serious hurdle for ribosome scanning and this could suggest another mechanism for translation initiation. Such a hypothesis is also strengthened by the fact that the AUG start codon is located at the vicinity and downstream of a four-way junction structure (Figure 2), a situation that is reminiscent of similar motifs found in some viral IRESes [49,82,83,84,85]. Last but not least, the SARS-CoV-2 5′UTR also contains an uORF starting at AUG^107^, which is located within the SL4 (Figure 2) and seems to be used as an initiation site, as suggested by ribosome profiling analyses in infected cells [72,86]. Interestingly and curiously, ribosome profiling analyses have also highlighted other uORFs initiating at near-cognate initiation codons [72,86]. One is of particular interest because it is located on a CUG codon at position 59 just 10 nt. upstream of the TRS leader, which means that it will be present on all the sgRNAs. However, one should bear in mind that ribosome occupancy observed in ribosome profiling analysis remains only indicative, as it is often difficult to distinguish between a real TIS and the pausing of the 48S preinitiation complex [72].

All these unusual structural features can have a great impact on the translational efficiency of the genomic and sub-genomic SARS-CoV-2 RNAs and this set the rationale for this study. By using a series of in vitro and ex vivo functional translational assays coupled with specific inhibitors of the cap-dependent mechanism, we have investigated and compared translation of all SARS-CoV-2 transcripts. Our data show some unexpected characteristics of the genomic RNA, which is extremely efficient to recruit ribosomes for protein synthesis in all systems assayed despite its length, complex RNA structure and the presence of a short upstream reading frame; these translational features are conserved with the SARS-CoV-1 but not with MERS-CoV.

## 2. Materials and Methods

### 2.1. DNA Constructs

The pGlo-Renilla, pEMCV-Renilla, pHCV-Renilla and pMM1-L-protease constructs have been previously described [87,88,89]. The leader sequence and the 5′UTR (5′UTR^ORF1ab^) of SARS-CoV-1 (NC_004718.3), MERS-CoV (NC_038294) and SARS-CoV-2 (NC_045512.2) gRNA have been generated by consecutive PCRs with specific primers (Appendix A). The resulting PCR products were digested with the BamHI restriction enzyme and inserted into the pRenilla vector between the BglII and BamHI restriction sites. The 5′UTRs of each SARS-CoV-2 sgRNA associated to the coding region of the Renilla luciferase were produced by consecutive PCRs with specific primers (Appendix A), digested with BamHI and EcoRV restrictions enzymes and inserted into the pRenilla vector between the BglII and EcoRV restriction sites. Mutation of the AUG^107^ and AUG^158^ in the SARS-CoV-2 and MERS-CoV gRNA, respectively, and the mutation of the UAA triplet at position 133 in SARS-CoV-2 have been performed by consecutive PCRs with specific primers (Appendix A) to generate the mut-AUG^107^, mut-AUG^158^ and uORF-phase constructs. The resulting PCR products were digested with the BamHI restriction enzyme and inserted into the pRenilla vector between the BglII and BamHI restriction sites. All DNA constructs were verified by DNA sequencing.

### 2.2. In Vitro Transcription

The four combinations—capped/polyadenylated (+/+), capped/non-polyadenylated (+/−), uncapped/polyadenylated (−/+) and uncapped/non-polyadenylated (−/−)—were obtained by in vitro transcription using the linear template of pRenilla vector by digestion either at the XbaI (non-polyadenylated RNAs) or at the EcoRI restriction site (polyadenylated RNAs). RNA transcription was performed from 1 μg of linear DNA template in transcription buffer (40 mM Tris–HCl (pH 7.5), 6 mM MgCl_2_, 2 mM spermidine and 10 mM NaCl) in the presence of 40U of T7 RNA polymerase (Promega, Madison, WI, USA), 40 U of RNAsin (Promega), 1.6 mM of each ribonucleotide triphosphate and 10 mM DTT. For capped mRNAs, the rGTP concentration was reduced to 0.32 mM, and m7-GpppG cap analog (New England Biolabs) was added at a concentration of 1.28 mM. The transcription reaction was carried out at 37 °C for 1 h 30 min, then treated for 30 min with the RQ1 DNAse (Promega). mRNAs were isolated using NucleoSpin RNA XS purification columns (Macherey Nagel, Düren, Germany). The integrity of RNA was checked by electrophoresis on non-denaturing 1% agarose gel and the concentration was determined by reading the absorbance using Nanodrop technology.

### 2.3. In-Vitro Translation Assays

Nuclease-treated Flexi Rabbit Reticulocytes lysate (RRL) or untreated RRL (Promega), supplemented as previously described [90,91], were programmed with 5 or 10 fmol of in vitro synthetized mRNAs in the presence of 75 mM KCl, 0.5 mM MgOAc, 2 mM DDT and 20µM of each amino acid for 30 min at 30 °C. For radioactive labeling, 0.25 mCi/mL of [^35^S] methionine was added to the reaction mix. Translation products were then stopped in either Renilla lysis-juice buffer (PJK Biotech, Kleinblittersdorf, Germany) for luciferase assay or in Laemmli sample buffer (4% SDS, 10% β-mercaptoethanol, 20% glycerol, 0.004% bromophenol Blue, 0.125 M Tris-HCl pH 6.8) and resolved by 12% SDS-PAGE. The gel was dried and quantified with a Molecular Dynamics PhosphoImager.

The L-protease from Foot-and-Mouth Disease Virus (FMDV) was prepared in RRL as previously described [87]. L-protease (0.2 µL and 0.4 μL), hippuristanol (0.15, 0.3 and 0.6 µM) and the cap analog (1 µM) were incubated 10 min in the RRL before RNA addition.

### 2.4. Renilla Luciferase Activity

Renilla luciferase activity was measured using the Renilla-Juice luciferase assay (PJK GmbH, Kleinblittersdorf, Germany) following the manufacturer instructions on a LUMIstar apparatus (BMG LABTECH, Champigny-sur-Marne, France) or on a Mithras LB940 (Berthold, Thoiry, France).

### 2.5. Western-Blot against eIF4G

Proteins from mock or L-protease-treated RRL extracts were separated by a 7.5% SDS-PAGE. The proteins were transferred onto polyvinylidene difluoride membranes (Boehringer Mannheim, Ingelheim am Rhein, Germany). Blots were incubated first with antibody against eIF4G (#2617 Cell Signaling Technology, Danvers, MA, USA) for 16 h at 4 °C, then with an anti-Rabbit IgG–Peroxidase antibody (Sigma Aldrich, Saint-Louis, MO, USA) for 2 h. The chemiluminescence signal was detected using an Pierce ™ ECL reagent (Thermofisher, Waltham, MA, USA) in the Chemidoc Imager (Biorad, Hercules, CA, USA).

### 2.6. Cell Culture and RNA Transfection

Jurkat T-cell line, obtained from ATCC, was cultured in RPMI-1640 medium (Life Technologies), supplemented with 10% fetal calf serum, penicillin, streptomycin, Hepes pH 7.5, sodium pyruvate and L-glutamine. Cells were incubated at 37 °C in a 5% CO_2_ atmosphere.

For RNA transfection, 5.10(5) cells were resuspended in 10µL of buffer R (Life technology) in the presence of 500 fmol of capped and polyadenylated mRNAs, and immediately transfected using the Neon transfection kit (Life technology, Carlsbad, CA, USA) according the manufacturer instructions. Cells were incubated in RPMI medium for 90 min at 37 °C, washed twice in PBS and split in two pools to measure luciferase activities and to extract cytoplasmic RNAs.

### 2.7. RNA Extraction and RT-qPCR

RNAs were isolated using the NucleoSpin RNA extraction kit according the manufacturer instructions (Macherey-Nagel, Düren, Germany) and quantified using a nanodrop 2000 spectrophotometer (Thermofisher, Waltham, MA, USA). Reverse transcription of 250 ng of cytoplasmic RNAs was performed using qScript kit (Quanta Bio, Beverly, MA, USA). mRNA quantification was performed by quantitative PCR with the ONEGreen^®^ FAST QPCR PREMIX (Ozyme, Saint-Cyr-l’École, France) according to the manufacturer instructions. Renilla luciferase (Renilla forward TGGACAATAACTTCTTCGTGAAAAC/ Renilla reverse GCTGCAAATTCTTCTGGTTCTAA) was amplified in parallel with the endogenous housekeeping gene glyceraldehyde 3-phosphate dehydrogenase (GAPDH) (GAPDH forward (Human) CGACAGTCAGCCGCATCTT/ GAPDH reverse (Human) CCCCATGGTGTCTGAGCG). The relative copy numbers of Renilla cDNAs were compared to GAPDH using x–Δ^Ct^ (where x corresponds to the experimentally calculated amplification efficiency of each primer couple).

### 2.8. Sequence Alignment

The 5′UTRs of the SARS-CoV-1 (NC_004718.3), MERS-CoV (NC_038294) and SARS-CoV-2 (NC_045512.2) gRNAs were analyzed with a Multiple Sequence Comparison by Log-Expectation (MUSCLE) using the default parameters [92].

## 3. Results

### 3.1. Translation Initiation Efficiency Mediated by the SARS-CoV-2 5′UTRs

During infection, SARS-CoV-2 generates at least 10 different mRNAs that harbor unique 5′UTRs in terms of length, structure and context of their AUG initiation site. In order to investigate further the role of these 5′UTRs in translational control, we have cloned each of the viral 5′UTR upstream of a reporter gene coding for the Renilla luciferase into the pRenilla vector [88,89,93] (Figure 3A). The nucleotide triplet downstream to the TIS has been conserved to maintain the SARS-CoV-2 original Kozak context. As a control, we have used the 5′UTR of the globin mRNA, which is known to promote translation initiation in a very efficient manner. Capped and polyadenylated RNAs have been produced in vitro to program the rabbit reticulocytes lysate (RRL) in the presence of S^35^-methionine. Protein production was quantified on a phosphoImager after an SDS-PAGE resolution (Figure 3B). For most of the transcripts, expression was as efficient as that generated by the 50 nt. of the β-globin 5′UTR (NC_000011.10; Figure 3B compare lanes 4–7, 9 and 11 with lane 2). Interestingly, the 5′UTRs derived from ORF1-ab, ORF-6 and ORF-7b promoted a higher level of expression than from the globin mRNA reporter gene (Figure 3B lanes 3, 8 and 10); this was not expected considering that these viral 5′UTRs are the longest and the most structured of the series (Figure 2 and Table 1). We were also surprised to find that translation of ORF-3a, ORF-S, ORF-E and ORF-6 was still very efficient despite the fact that their AUG start codons are in a poor Kozak context (with a cytosine in -3 and/or an uracil in +4; Table 1).

Quantification of protein production was also performed by measurement of the luciferase activity (Figure 3C). It should be remembered that the first amino acid downstream to the methionine codon was variable for each construct as we conserved the Kozak context of the original TIS (Appendix A). Thus, this N-terminal modification could have an impact on the activity of the luciferase; the same samples have also been quantified by luciferase assay. As the results obtained by reading of the luciferase assays were consistent with S^35^-methionine quantification (compare Figure 3B and Figure 3C), we assumed that this N-terminal modification did not impact significantly the enzymatic activity of luciferase and the latter has been used as a readout for protein production for the remainder of this study.

Next, we translated the different mRNAs into the untreated RRL, which represents an in vitro competitive translation system due to the presence of endogenous mRNAs that compete for the translational apparatus [89]. In such a system, translation initiation mediated by the majority of the SARS-CoV-2 5′UTRs was further enhanced in comparison to the expression mediated by the globin 5′UTR (Figure 3D). Once again, and consistent with the RRL results, translation from the ORF-1ab 5′UTR was the most efficient. This very high level of translation of viral RNAs in the untreated RRL suggests that they have the intrinsic ability to recruit ribosomes in a competitive environment. To go further, the mRNAs have been electroporated into cells and the protein expression level has been measured by luciferase assay and normalized to the quantity of RNA by RT-qPCR. The results showed in Figure 3E are consistent with the data obtained in the RRL systems. Taken together, our data show that the 5′UTRs of the SARS-CoV-2 have the ability to promote a high level of translation both in vitro and ex-vivo, with the 5′UTR^ORF1ab^ (gRNA) sequence being the most efficient of all.

### 3.2. Translation Efficiency from MERS-CoV gRNA 5′UTR Differs from SARS-CoV gRNA 5′UTRs

We next wondered whether a high level of translation could also be observed with RNAs derived from closed relative strains, i.e., the SARS-CoV-1 [94,95,96] and the MERS-CoV [97,98,99]. The 5′UTR of the SARS-CoV-1 gRNA is 264 nt. long and shares 90% of its identity with the sequence of the SARS-CoV-2 (Figure 4A and Appendix A). Once more, the global RNA structure of the 5′UTR is also conserved [73,74,77], which could indicate that expression initiated from the 5′UTR of the gRNA from both SARS-CoV strains could be similar. In contrast, the MERS-CoV gRNA’s 5′UTR is slightly longer (277 nt.) and more distant from SARS-CoV sequences as it shares only ~60% of its identity with type 1 and type 2 (Figure 4A and Appendix A), which could also suggest a divergence in translational efficiency. To directly test this hypothesis, we have used the pRenilla reporter gene harboring either the leader sequence, which represents the shorter 5′UTR (named leader), or the 5′UTR of the SARS- and the MERS-CoV gRNA (named 5′UTR^ORF1ab^ in the rest of the manuscript) (Figure 4B). Capped and polyadenylated mRNAs were generated by in vitro transcription and translated in the RRL (Figure 4C) or electroporated into Jurkat cells (Figure 4D). The products of expression were quantified by luciferase assays and normalized to the expression mediated by the globin 5′UTR, which served as a positive control. Consistent with our previous results, the 5′UTR^ORF1ab^ of SARS-CoV-2 gRNA promoted strong protein production (Figure 4C,D). As expected, a similar expression profile was observed when translation was driven by the 5′UTR^ORF1ab^ of the SARS-CoV-1, both in the RRL and in cells (Figure 4C,D), confirming that the translational characteristics are well conserved between the two strains. In contrast, although the leader sequence of MERS-CoV was also efficient to drive translation in the RRL, it still remained lower than that of the SARS-CoV transcripts (Figure 4C). This difference was even sharper with the 5′UTR^ORF1ab^ of MERS-CoV, which showed a decrease of up to 30% when compared to globin expression (Figure 4C). These differences were also observed when the reporter mRNAs were transfected in cell lines (Figure 4D). Our data show that the translation characteristics of the 5′UTR^ORF1ab^ are conserved between the two SARS-CoV strains but differ with MERS-CoV. This indicates that divergences in sequence conservation are also observed in terms of translational efficiency.

### 3.3. SARS-CoV-2 Translation Initiation Requires the Cap Structure and the Components of eIF4F Complex

Data obtained so far have pointed out that translation of the gRNA of the SARS-CoV-2 is very efficient, both in vitro and ex vivo, but the molecular mechanism by which this occurs remains to be determined. Surprisingly, the gRNA is the longest and most structured transcript, which should, a priori, represent a physical barrier for linear ribosomal scanning. On the other hand, the viral gRNA is capped and polyadenylated, which is a strong argument for a cap-dependent mechanism of translation initiation. This is also in agreement with previous studies showing that addition of cap analog to the RRL resulted in a strong reduction of translation driven by the 5′UTR of the bovine CoV strain [100,101]. Furthermore, the destabilization of the eIF4F complex also results in the inhibition of viral replication of MERS-CoV and HCoV-229E [102,103]. Although these studies suggest a cap-dependent mechanism of translation for coronavirus mRNAs, this has not been experimentally established. On the other hand, the structural features of the 5′UTR^ORF1ab^ (gRNA), as well as the presence of a uORF and an AUG start codon located in a four-way junction RNA structure, could suggest the presence of an active IRES motif within the 5′UTR [49,82,83,84,85]. Therefore, in order to distinguish between these two possibilities, we decided to further investigate the molecular mechanism by which translation initiation occurs on the SARS-CoV-2 gRNA.

We first looked at the effects of both the cap and the poly(A) tail structures in the translation of SARS-CoV-2 mRNAs. In order to avoid the bias of RNA stability caused by the absence of both structures, we performed our study in in vitro translation systems. For this, we produced different combinations of capped, uncapped, polyadenylated and non-polyadenylated transcripts harboring the 5′UTR of the globin, the 5′UTR^ORF1ab^ or the leader sequence of SARS-CoV-2. Both RRL (Figure 5A) and untreated RRL (Figure 5B) were programmed for 30 min and expression of the Renilla was quantified by luciferase assays. As expected, the synergic effect of the combination of the cap structure and the poly(A) tail was observed with the RNA control (globin 5′UTR). Enhancement of translation provided by these two elements was even amplified in the competitive untreated RRL, which is an experimental system known to recapitulate faithfully cap/poly(A) synergy [89]. Expression initiated by the leader sequence or the 5′UTR^ORF1ab^ was highly dependent on the presence of the cap structure with a 10-fold stimulation over the uncapped transcripts (Figure 5B). In contrast, we were surprised to find that the addition of the poly(A) tail on the SARS-CoV-2 mRNAs did not add any supplementary translational advantage over the non-polyadenylated mRNAs. This shows an unambiguous requirement for the m^7^GTP cap structure but no significant effect of the poly(A) tail. In addition, this denotes a major difference with canonical globin translation regarding the requirement for the 3′ poly(A) tail.

We decided to continue our investigation to look at the role of the initiation factors that are required for SARS-CoV-2 translation with a particular focus on the cap-binding complex eIF4F (Figure 6A). This complex is formed by the association of three initiation factors: eIF4E, eIF4A and eIF4G and its role is to bind both to the m^7^GTP cap structure and to the ribosome [2]. Addition of cap analog to the RRL chelates eIF4F by competitive binding and sequesters it from binding to the 5′ end of the mRNAs (Figure 6A) [104]. As expected, the addition of the cap analog in our translation assay resulted in a strong repression of globin translation (Figure 6B) and a similar effect was observed on both the 5′UTR^ORF1ab^ and leader of the SARS-CoV-2 transcripts, confirming that the eIF4F complex is needed for their expression.

Next, in order to distinguish whether the whole eIF4F, or parts of it, are needed, we decided to specifically focus on the independent role of both eIF4A and eIF4G, which are core subunits of eIF4F. eIF4G can be targeted by viral proteases and eIF4A can be specifically inhibited by some chemicals (Figure 6A). We first studied the dependence on eIF4G by using the L protease from the foot-and-mouth-disease virus [105,106]. This enzyme cleaves eIF4G to yield a N-terminal domain able to bind to eIF4E and a C-terminal domain (called p100), which harbors the interacting domains with eIF4A and eIF3 [107]. Functionally, the cleavage of eIF4G inhibits cap-dependent transcripts but does not affect, and even stimulates, translation of mRNAs bearing an IRES [108]. We have added in-vitro-translated L protease to the untreated RRL that resulted in the cleavage of eIF4G (Figure 6C) and we observed different effects on the reporter constructs assayed. As expected, cap-dependent translation driven by globin 5′UTR was inhibited (Figure 6D), whereas the encephalomyocarditis virus (EMCV) IRES-mediated protein synthesis was slightly stimulated (Figure 6E) as previously described [87]. Cleavage of eIF4G resulted in the inhibition of expression from both SARS-CoV-2 5′UTR^ORF1ab^ and leader sequence mRNAs (Figure 6F) and to a similar magnitude as that for globin, suggesting that translation initiation from SARS-CoV-2 5′UTRs depends on a mechanism that requires the integrity of eIF4G.

Finally, we evaluated the requirement for the DEAD box RNA helicase eIF4A, which is the third partner of the eIF4F complex and is involved in RNA unwinding and scanning [109]. Indeed, chemical and enzymatic probing analysis of the SARS-CoV-2 5′UTR^ORF1ab^ [73,74] has revealed a high level of RNA secondary structures (Figure 2) that may interfere with ribosome scanning. To test this hypothesis, we used a natural chemical inhibitor of the eIF4A RNA-binding activity called hippuristanol (Figure 6A) [110]. Addition of increasing amounts of hippuristanol in the RRL inhibited globin cap-dependent translation in a dose-dependent manner, whereas it did not interfere with expression of a mRNA driven by the Hepatitis C Virus (HCV) IRES (Figure 6G), which does not require eIF4A [111]. The RRL was then programmed with reporter genes containing SARS-CoV-2 5′UTR^ORF1ab^ or leader sequence RNA constructs and this showed that initiation on both of these RNAs was repressed upon addition of hippuristanol (Figure 6G) and the extent of translation repression was comparable to that obtained for globin mRNAs.

Taken together, our data indicate that translation from the SARS-CoV-2 5′UTR^ORF1ab^ requires the activity of each component of the eIF4F complex, namely eIF4E, eIF4G and eIF4A, which is consistent with a cap-dependent initiation mechanism.

### 3.4. uORF Downregulates the Translation Efficiency of gRNA

The 5′UTR^ORF1ab^ of the SARS-CoV-2 (gRNA) harbors a potential uORF of nine amino acids that starts at the AUG^107^ triplet and finishes at the UAA triplet located at position 134 (Figure 2 and Figure 7A). uORFs are *cis*-acting elements located in the 5′UTR that impact on the translation efficiency of the mRNA. When these uORFs are translated by incoming ribosomes from the 5′ end, this often results in the decrease of translation initiation from the main downstream ORF [27,112,113,114,115]. However, in some particular cases, the presence of uORFs can also act as enhancers of protein production, as it was described with some stress-related mRNAs [29,114,116,117].

In the case of the SARS-CoV-2 gRNA, ribosomal profiling data have suggested that some ribosomes recognize and pause at the AUG^107^ codon [86]. However, formal proof that this uORF is translated and the level at which it is expressed remain to be determined. To test this hypothesis, we introduced two mutations in the pSARS-CoV-2 5′UTR^ORF1ab^-Renilla to change the AUG^107^ triplet into an UAA triplet (Mut-AUG^107^) and to delete the stop codon (UAA^134^) in order to align AUG^107^ in the same reading frame as the Renilla ORF (uORF phase). The resulting RNAs have been synthesized by in vitro transcription and translated into the RRL in the presence of S^35^-methionine. Data are presented in Figure 7 and show that mutation of the AUG^107^ triplet resulted in a slight stimulation of expression of the main ORF (Renilla) (Figure 7B lane 3), confirming that this AUG triplet was recognized as an initiation site by the scanning ribosome. This was confirmed by the second mutation (deletion of the UAA), which resulted in an increased expression of Renilla (Figure 7B lane 4, denoted by the star). Altogether, our data unambiguously show that the uORF located in the 5′UTR^ORF1ab^ was translated in the RRL, but the effect on protein expression, i.e., both production of the short peptide and increased production of Renilla, remained relatively modest. In order to better appreciate the magnitude of this effect, we have performed similar experiments on the 5′UTR^ORF1ab^ of the MERS-CoV gRNA as it also contains an uORF starting at the AUG^158^ triplet, which is located closer to the AUG start codon of ORF-1ab (Figure 7A). To evaluate its role on mRNA translation, we have also mutated the AUG^158^ triplet. The resulting mRNA was translated in the RRL and protein products were analyzed on SDS-PAGE and quantified with a phosphoImager. We observed that mutation of the AUG^158^ triplet induced a strong expression of the main ORF, which was stimulated by up to three times when compared to the parental 5′UTR (Figure 7C). This stimulation was also observed when the reporter mRNAs were transfected in cell lines (Figure 7D). These results confirmed important differences in translational control exerted by the uORFs in the coronavirus family with a moderate attenuation of translation in SARS-CoV-2 compared to a more pronounced impact on the MERS-CoV 5′UTR.

## 4. Discussion

Infection by SARS-CoV-2 generates a gRNA and at least nine sgRNAs, which all start with a 75 nt. common leader sequence [71]. Each of these transcripts harbor a unique 5′UTR, which is composed of the shared leader sequence followed by an RNA region that varies in length (between 0 to 190 nt. (Table 1)). Thus, the resulting viral 5′UTRs have different degrees of RNA structures and some of them contain *cis*-acting elements, such as an uORF, which is known to have an impact on translation initiation efficiency. This set the rationale for this study aimed at investigating the translational properties of all virally encoded mRNAs. To answer to this question, we showed that viral 5′UTRs show significant variations of expression although they all exhibit a high level of activity compared to globin 5′UTR control. Importantly, the high and relative levels of expression between viral mRNAs were maintained regardless of the experimental systems used (RRL, competitive RRL and cells) (Figure 3), with the 5′UTR^ORF1ab^ (gRNA) being the most efficient of all. This is rather surprising considering that the 5′UTR^ORF1ab^ is the longest of the series (265 nt.), with multiple stable hairpin RNA structures and, above all, the presence of the uORF (Figure 2). A priori, these structural features should represent hurdles for ribosomal binding and scanning and one would predict a poor level of protein expression. However, this is clearly not the case as expression was revealed to be very efficient and conserved between the two SARS-CoV strains (Figure 4). This suggested to us that the SARS-CoV-2 5′UTR^ORF1ab^ could use an alternative cap-independent mechanism; such an assumption was also strengthened by the presence of the AUG start codon in a four-way junction, which is a structural conformation that was previously described for HCV IRES [118]. This prompted us to look for the requirement of the viral gRNA in the initiation factors involved in cap-driven translation initiation. We showed that both the cap structure and the eIF4F complex are necessary to initiate translation, which strongly suggest a cap-dependent mechanism (Figure 5 and Figure 6). We can assume that, after ribosomal attachment to the cap structure, the PIC scans the viral 5′UTR to reach the initiation codon as eIF4A is required (Figure 6G) and, above all, the uORF is translated (Figure 7). However, we cannot rule out that ribosomal shunting could occur through the 5′UTR to bypass some hairpin RNA structures [119,120,121] and contribute to this high level of translation. This mechanism usually needs additional proteins [121] and this will require further studies to determine its implication in SARS-CoV-2 translation initiation.

In the canonical translation initiation mechanism, the interaction between the poly(A) binding protein (PABP) and eIF4G [122,123] brings the 5′- and 3′-ends of the mRNA together and forms a mRNA’s circularization called the closed-loop model [124,125], which promotes translation [89,126]. Interestingly, in our system, which is able to recapitulate the cap/poly(A) synergy [89], the poly(A) tail did not provide any translational advantage to the mRNAs that harbor the SARS-CoV-2 5′UTR^ORF1ab^ when compared to globin 5′UTR, which uses a strict canonical cap-dependent translation initiation mechanism (Figure 5). This particularity is quite unusual in the eukaryotic kingdom and this raises the question of whether this closed-loop conformation takes place on mRNAs that harbor the SARS-CoV-2 5′UTR. The poly(A) tail requirement should be further investigated in the future to determine if the PAPB is necessary to promote initiation and still interacts with eIF4G. To our knowledge, a neutral influence of the poly(A) tail, in translation, has never been reported before. Viral and cellular mRNAs without a poly(A) tail have been described but they use alternative and specific *cis*-acting elements to bring the 5′- and 3′-ends of the mRNA together [127,128,129,130,131,132]. Analyses of the formation of 48S PIC in RRL through a sucrose gradient have indicated that it is almost absent on mRNAs harboring the SARS-CoV-2 5′UTR but not with the EMCV IRES [80], suggesting that the initiation process on SARS-CoV-2 is not canonical. In SARS-CoV-2 infected cells, the size of the poly(A) tail is quite short (47 nt. for median length) and is longer on the gRNA than on the sgRNAs [71]. Interestingly, during bovine CoV infection, the length of the poly(A) tail changes from ~45 nt. immediately after virus entry to ~65 nt. at 6 to 9 h post-infection [133]. These quick modifications of the length are reminiscent of the translational control that takes place during early oocyte/embryo development [134]. Although we did not notice a requirement for the poly(A) tail in SARS-CoV-2 translation in the RRL, this does not exclude that the poly(A) tail of SARS-CoV-2 mRNAs may regulate translation efficiency, mRNA stability and/or mRNA localization in infected cells.

The exact mechanism by which the SARS-CoV-2 5′UTRs drive the high levels of expression is puzzling considering the structural features of the gRNA; nevertheless, this characteristic could turn out to be a real benefit for viral replication in infected cells. Indeed, in the early phases of infection, the viral nsp1 interacts with the 18S ribosomal RNA in the mRNA entry channel of the ribosome and imposes a blockage by steric hindrance, leading to a global inhibition of mRNA translation [135,136,137]. However, viral mRNAs can escape this nsp1-mediated repression via a complex interplay between the leader sequence, especially the SL1 RNA structure [78,80,81,137], which is able to interact with nsp1 [80,138]. It has been shown that, at low concentrations, nsp1 is able to stimulate the expression of mRNA reporters that harbor the SL1 sequence in cells [81] but not in the RRL [80], which is not a competitive system. This, along with our data, suggests that SARS viral RNAs have the ability to compete very efficiently with cellular transcripts for the translational apparatus. Thus, it would be very interesting to assay the role of nsp1 in the untreated RRL as well as in infected cells.

Finally, the expression of the gRNA is also regulated by an uORF localized within its 5′UTR, some 120 nucleotides upstream from the authentic AUG initiation codon. We first showed that this uORF was recognized by scanning ribosomes and translated into a short peptide whose role remains to be determined. In addition, mutation of the start AUG codon of the uORF induced only a mild stimulation of the expression of the main reading frame. This contrasts with the situation obtained with the MERS-CoV, in which we showed that this uORF exerts a strong control on translation (Figure 7C,D). This is puzzling as the uORF is conserved, suggesting its importance in the control and regulation of protein synthesis despite having different effects on the flux of incoming ribosomes.

Genome-wide sequencing of 5′UTR has revealed that uORFs are much more pervasive in the genome than initially described, with an occurrence of about 50% in mammalian mRNAs [139,140]. These uORFs can be located within non-coding RNAs, within the 5′ and 3′ UTRs or they can also be overlapping with the main ORF, either in-frame or out-of-frame with the main protein product [141]. According to their localization, these uORFs regulate gene expression in at least three different ways: (i) by imposing a sharp translational control on the main ORF; (ii) by producing functional micro-peptides and (iii) generating truncated/extended isoforms of the main protein product.

In the case of the uORF present in the SARS/MERS coronavirus, it is clearly located within the 5′UTR^ORF1ab^ and encodes a putative short peptide of 9–11 amino acids, which does not appear conserved amongst the different members (Appendix A). Thus, it suggests that this uORF regulates expression of the main protein product by throttling down the flux of incoming ribosomes. This can happen by either stalling the 80S ribosomal complex that forms on the encoded short reading frame, thus creating a roadblock for scanning ribosomes or, alternatively, the short reading frame can also be a waylay for scanning pre-initiation complexes [22]. As the same conserved uORF has a different impact on expression from the SARS-CoV-1 and -2 compared to the MERS-CoV, this could reflect the fact that two distinct mechanisms are at play. This is an important point that will be addressed in the future. In any case, this uORF must be a key regulator of viral translation and replication as it is widely conserved. Indeed, this uORF is highly conserved from different coronaviruses [142] and recent ribosome profiling analyses have confirmed that its AUG initiation codon is recognized by ribosomal complexes during both Mouse Hepatitis Virus (MHV) and SARS-CoV-2 infection [86,143]. One study has focused on the physiological function of the peptide issued by this uORF in MHV infection. For that, the authors mutated the AUG-initiated uORF in the viral genome, showing that the virus was viable in cell culture. However, serial passage of this mutant variant resulted in a reversion to an intact uORF, indicating that this uORF must play a beneficial role in virus survival in cell culture [142].

To summarize, our data strongly suggest that translation of the SARS-CoV-2 mRNAs involves an unconventional cap-dependent mechanism that involves *cis*-acting RNA elements such as the structure of the leader, the length of the poly(A) tail and the role of the uORF. Taken together, these elements must probably functionally interact with each other in order to regulate the flow of ribosomes on the viral mRNAs.

## Figures and Tables

**Figure 1 viruses-14-01505-f001:**
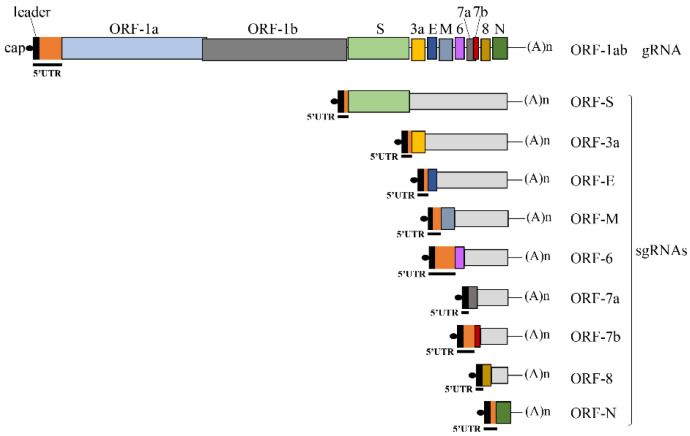
Scheme of the SARS-CoV-2 gRNA and sgRNAs. The cap and poly(A) tail structures are indicated. The 5′UTR of each viral mRNA is underlined and is composed of the common leader sequence (black square) and the variable region (orange square). The other different colors correspond to the different ORFs as indicated on the top.

**Figure 2 viruses-14-01505-f002:**
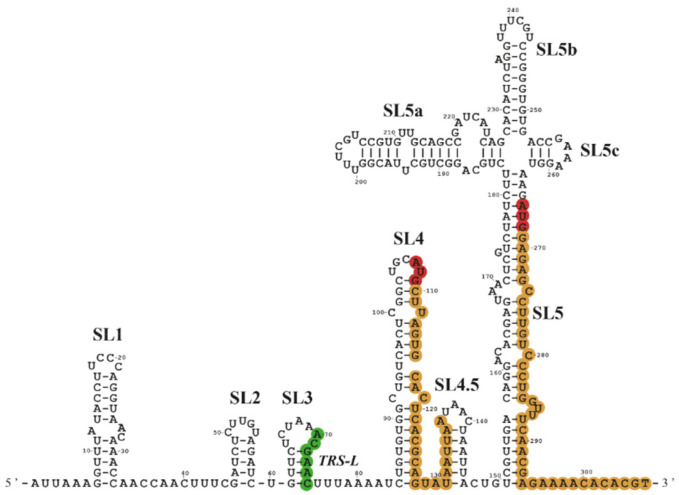
Modelization of the structure of the SARS-CoV-2 gRNA 5′UTR. This model, adapted from Miao et al. (2021) [73], shows the different stem-loop (SL) RNA structures. Position of the AUG codons of uORF and main ORF are highlighted in red, the coding region in orange and the TRS-L in green.

**Figure 3 viruses-14-01505-f003:**
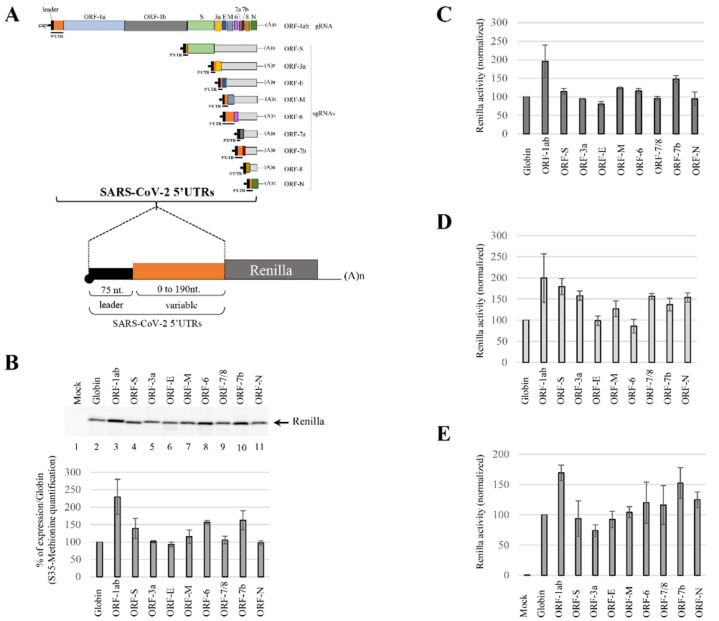
Translation efficiency of the SARS-CoV-2 5′UTRs. (**A**) Scheme of the gRNA and the different sgRNAs, with the leader sequence in black and the variable region in orange, which represent each 5′UTR (this panel was obtained from Figure 1). The sequences corresponding to each 5′UTR have been cloned upstream of the Renilla ORF. (**B**,**C**) RRL and (**D**) untreated RRL have been programmed with capped and polyadenylated mRNAs harboring specifically the 5′UTR of the ORF1ab, ORF-S, ORF-3a, ORF-E, ORF-M, ORF-6, ORF-7a, ORF-7b, ORF-8 or ORF-N. The resulting protein products have been resolved on (**B**) an SDS-PAGE and quantified with a phosphoImager, or (**C**,**D**) analyzed by luciferase assay. (**E**) Jurkat cells were electroporated with capped and polyadenylated mRNAs as indicated. After 90 min of incubation, the cellular extracts were used for RT-qPCR and luciferase assay. All the results are normalized to the globin 5′UTR expression (set at 100%). Values are the mean (+/− S.D.) for three independent experiments (panels (**B**,**C**)) or (+/− S.E.M.) for four independent experiments (panels (**D**,**E**)).

**Figure 4 viruses-14-01505-f004:**
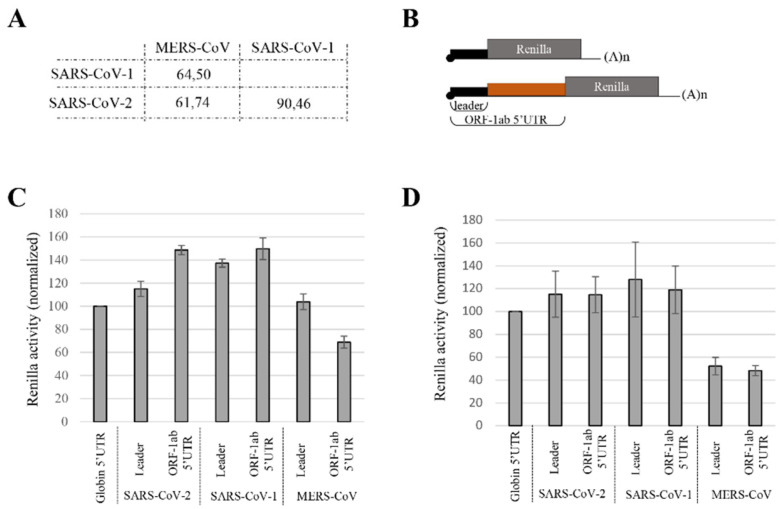
Divergence in translation efficiency between the 5′UTRs of MERS-CoV and SARS-CoV gRNA. (**A**) Percentage of identity between the 5′UTRs of MERS-CoV, SARS-CoV-1 and SARS-CoV-2 gRNAs resulting from a MUSCLE alignment (Appendix A). (**B**) Scheme of capped and polyadenylated mRNAs coding for the Renilla reporter gene with the leader or the 5′UTR^ORF1ab^ from SARS-CoV-1/2 and MERS-CoV. (**C**) RRL was programmed with the corresponding in vitro transcribed reporter mRNAs. Expression products were quantified by luciferase assays and normalized to globin 5′UTR expression (set at 100%). (**D**) Jurkat cells were electroporated with capped and polyadenylated mRNAs as indicated. After 90 min of incubation, the cellular extracts were used for luciferase assay and RT-qPCR. All the results are normalized to the globin 5′UTR expression (set at 100%). Values shown are the mean (+/− S.E.M.) for 6 (**C**) and 5 (**D**) independent experiments.

**Figure 5 viruses-14-01505-f005:**
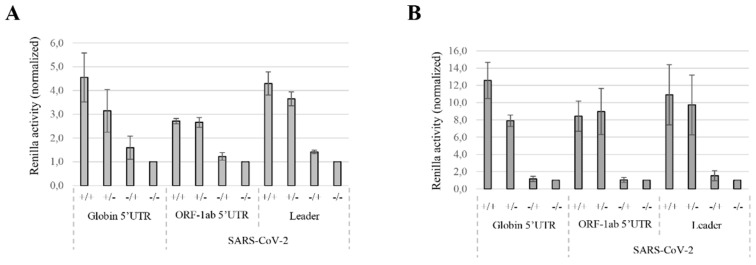
The effects of the cap and poly(A) tail on viral translation. Capped/polyadenylated (+/+), capped/non-polyadenylated (+/−), uncapped/polyadenylated (−/+) or uncapped/non-polyadenylated (−/−) Globin Renilla, 5′UTR^ORF1ab^ Renilla and leader Renilla RNAs were translated in (**A**) the RRL or in (**B**) the untreated RRL. Translational products were analyzed by luciferase assays and normalized to the (−/−) condition (arbitrary set at 1). Values shown are the mean +/− S.D. for three independent experiments.

**Figure 6 viruses-14-01505-f006:**
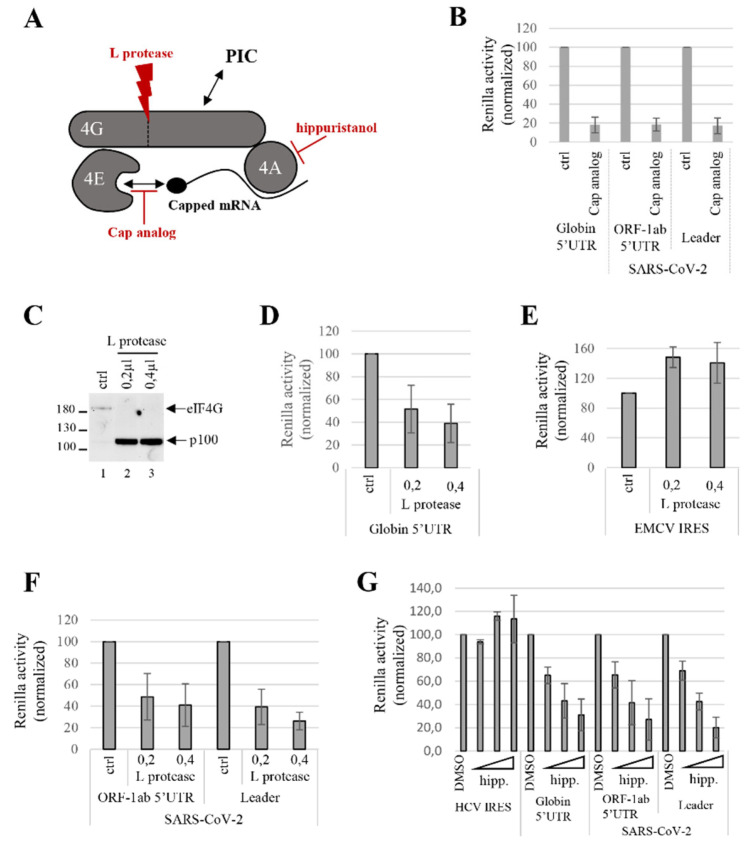
Requirement for the eIF4F complex. (**A**) Schematic cartoon of the eIF4F complex, which is composed of eIF4E, eIF4G and eIF4A. The interactions with the PIC, cap analog, L protease and hippuristanol are depicted on the figure. (**B**) RRL was treated with cap analog (1 µM) or with water (ctrl) and then programmed with capped and polyadenylated globin-Renilla and SARS-CoV-2 leader or 5′UTR^ORF1ab^-Renilla mRNAs. Products of expression were quantified by luciferase assays and normalized to the control condition (set at 100%). (**C**–**F**) Untreated RRL was pre-incubated alone (ctrl) or in presence of different amounts of L protease (µL) as indicated, and then programmed with (**D**) globin-Renilla, (**E**) EMCV IRES-Renilla, (**F**) SARS-CoV-2 leader or 5′UTR^ORF1ab^-Renilla mRNAs. Translation products were (**C**) resolved by a 7.5% SDS-PAGE and then transferred to a PVDF membrane for Western blotting analysis against eIF4G, or (**D**–**F**) analyzed by luciferase assay and normalized to the control condition (set at 100%). (**G**) RRL pre-treated with DMSO or increasing amounts (µM) of hippuristanol (hipp.) was programmed for 30 min with globin-Renilla, HCV-IRES Renilla, SARS-CoV-2 leader or ORF-1ab 5′UTR-Renilla mRNAs. Expression products were analyzed by luciferase assay and normalized to the control condition (set at 100%). Values shown are the mean (+/− S.D.) for three independent experiments.

**Figure 7 viruses-14-01505-f007:**
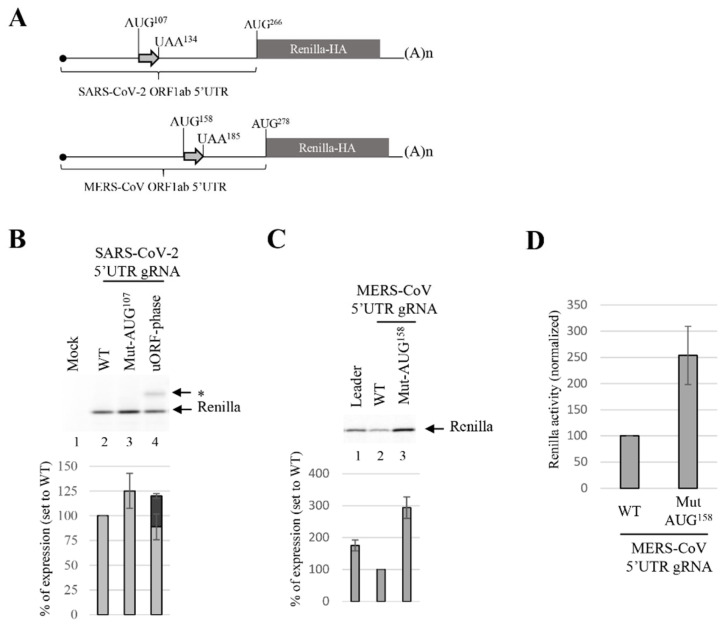
The 5′UTR uORF differentially regulates translation from SARS-CoV-2 and MERS-CoV. (**A**) Scheme of the reporter mRNA with the 5′UTR^ORF1ab^ of SARS-CoV-2 or MERS-CoV gRNA with their respective uORF and positions. (**B**) Capped and polyadenylated mRNA harboring the SARS-CoV-2 5′UTR gRNA WT or the Mut-AUG^107^ or uORF-phase mutations were translated in the RRL. Translation products were resolved by 12% SDS-PAGE and quantified with a phosphoImager. (**C**) Capped and polyadenylated mRNA harboring the MERS-CoV 5′UTR gRNA WT or the Mut-AUG^158^ mutation, were translated in the RRL. Translation products were resolved by 12% SDS-PAGE and quantified with a phosphoImager. (**D**) Jurkat cells were electroporated with capped and polyadenylated mRNAs as indicated. After 90 min of incubation, the cellular extracts were used for luciferase assay and RT-qPCR. Values shown are the mean (+/− S.D.) for three independent experiments.

**Table 1 viruses-14-01505-t001:** Length, Kozak context and %GC of the SARS-CoV-2 gRNA and sgRNA 5′UTRs.

5′UTR	Length (nt.)	Kozak Context ^1^	%GC
ORF-1ab	265	GGU AAG **AUG** GAG	45
ORF-S	76	CGA ACA **AUG** UUU	37
ORF-3a	77	GAA CUU **AUG** GAU	36
ORF-E	77	GAA CUU **AUG** UAC	36
ORF-M	119	UUA GCC **AUG** GCA	31
ORF-6	230	CAA CAG **AUG** UUU	41
ORF-7a	75	ACG AAC **AUG** AAA	37
ORF-7b	151	GAC AGA **AUG** AUU	34
ORF-8	75	ACG AAC **AUG** AAA	37
ORF-N	83	ACU AAA **AUG** UCU	35

^1^ The AUG codons are represented in bold and the nucleotides at positions −3 and +4 are underlined.

## Data Availability

Requests for further information about resources, reagents, and data availability should be directed to the corresponding author.

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
