# Peer review of "Translation of SARS-CoV-2 gRNA Is Extremely Efficient and Competitive despite a High Degree of Secondary Structures and the Presence of an uORF"

_viruses, 2022, doi:10.3390/v14071505_

Round 1

Reviewer 1 Report

This paper evaluated translation initiation efficiencies of different 5’ UTRs from SARS-CoV-2 sgRNAs. The authors attached these different 5’ UTRs to the same renilla luciferase sequence. It is indeed surprizing that the viral 5’ UTRs are as good as, or better than, the globin 5’ UTR. The paper is well written and contains significant results worth publishing.

There are a few things loose that should be revised.

1.       It is not clear if the 5’ UTR is taken from α globin or β globin. The globin gene name should be specified and the number of nucleotides in the 5’ UTR should be specified.

2.       It is not clear if the Renilla luciferase is the original from Renilla muelleri or have been modified in some way (The authors did mention their modification of the second codon). The actual coding sequences of Renilla luciferase should be included in the supplemental material. Alternative (if the modifications are few), one can have a short table highlighting the difference in sequences from the original coding sequence. That is, show the original sequence and highlight the modifications

3.       The last row in Table 1 “Leader” is not clear. Which sgRNA is it referring to?

Some statements are strong but not substantiated by their cited references. For example, “The positions at − 3 and the + 4 (according to the position of the AUG codon) are determinant because they directly interact with the 18S RNA of the 40S ribosomal subunit and eIF2 [17].” Reference [17] is a crosslinking experiment. If one does not know that -3 or +4 sites are important, then few researchers would attribute site importance to outcomes of crosslinking experiments. In fact, there is hardly any consistent support for the +4G site as important for translation initiation, although the -3R has been substantiated in many experiments. It has been argued before (PMID: 17285142) that the preponderance of +4G site is simply due to the following. First, more than half of the proteins undergo N-terminus methionine excision (NME). Second, NME requires a small and nonpolar amino acid (which typically means glycine or alanine). As glycine and alanine are encoded by GGN and GCN, we have AUG GGN or AUG GCN, i.e., a preponderance of +4G. One also notes that highly expressed genes do not exhibit a preference for +4G.

Some inferences are weak, e.g.,

“Clustal multiple sequence alignment of the different 5’UTRs (gRNA and sgRNAs) established from SARS-CoV-2 variants (alpha, beta, delta, gamma and omicron) showed a very high conservation of these sequences (Figure S1) suggesting that RNA cis-acting elements within the 5'UTR are of importance for either replication or translation or both”. This alignment means very little. The sequence variation is comparable to the rest of the SARS-CoV-2 sequences, i.e., the 5’ UTRs are NOT more conserved than other part of the viral genome (They are actually less conserved than the coding sequences) and do not suggest that “RNA cis-acting elements within the 5'UTR are of importance for either replication or translation or both”. By the way, MAFFT (PMID: 23329690) and MUSCLE (PMID: 15318951) are much better than Clustal.

The discussion of IRESes is also problematic. While the structures of viral IRESes are well documented to help with ribosome recruitment, Eukaryotic IRESes documented so far are generally devoid of secondary structures and they generally have much weaker IRES activities than those observed in viral IRESes. The lack of structure is interpreted to aid the exposure of the initiation AUG to facilitate translation initiation (PMID: 19125192; 17761883). The statement “Despite differences in length, sequence and structures, all IRESes have the characteristics to directly interact with some components of the 43S initiation complex to bring the ribosome close to the initiation codon [43, 46]” is not substantiated by the two cited references.

Author Response

Reviewer 1

We thank the reviewer for their insightful comments and have incorporated all the suggestions resulting in a much-improved manuscript.

Point 1: “It is not clear if the 5’ UTR is taken from α globin or β globin. The globin gene name should be specified and the number of nucleotides in the 5’ UTR should be specified.”

Response: The reviewer is right and this may be confusing for the reader. The study was performed with the 50 nucleotides of the β-globin 5’UTR (NC_000011.10). This is now mentioned in the section 3.1 as follows:

“For most of the transcripts, expression was as efficient as that generated by the 50 nt. of the β-globin 5’UTR (NC_000011.10; Figure 3B compare lanes 4-7, 9 and 11 with lane 2).”

Point 2: “It is not clear if the Renilla luciferase is the original from Renilla muelleri or have been modified in some way (The authors did mention their modification of the second codon). The actual coding sequences of Renilla luciferase should be included in the supplemental material. Alternative (if the modifications are few), one can have a short table highlighting the difference in sequences from the original coding sequence. That is, show the original sequence and highlight the modifications”

Response: The Renilla luciferase ORF originally provides from the pRL-CMV vector (Promega – ref: E2261). According to Promega, the pRL Vectors contain the cDNA encoding Renilla luciferase (Rluc) cloned from the anthozoan coelenterate Renilla reniformis. To help the reader to find this information and to understand the historic of the pRenilla construct, we added the two following references - Soto-Rifo et al 2007 PMID: 17881372 and Dizin et al 2006 PMID: 16782705 – in sections 2 (DNA constructs) and 3.1.

Concerning the N-terminal modification, to keep the nucleotide context of the initiation codon (figure 3) we had to modify the codon just downstream the AUG codon. As a consequence, only the nature of the second amino acid changes as follows:

MDTSKVYDPE – Globin

METSKVYDPE – ORF-1ab

MFTSKVYDPE – ORF-S / ORF-6

MDTSKVYDPE - ORF-3a

MYTSKVYDPE – ORF-E

MATSKVYDPE – ORF-M

MKTSKVYDPE – ORF-7a / ORF-8

MITSKVYDPE – ORF-7b

MSTSKVYDPE – ORF-N

The reader can easily find these modifications in the table 1 in which the nucleotide context of the initiation codon is indicated from the -6 to +6 nucleotide positions (according to the adenosine of the AUG codon (+1)). To help the reader and as suggested by the reviewer, these single N-terminal modifications are indicated in the supplemental figure S1.

Point 3: “The last row in Table 1 “Leader” is not clear. Which sgRNA is it referring to?”

Response: We thank the reviewer for this remark. The leader corresponds only to the first 75 nucleotides and may confuse the reader. This has been removed.

Point 4: “Some statements are strong but not substantiated by their cited references. For example, “The positions at − 3 and the + 4 (according to the position of the AUG codon) are determinant because they directly interact with the 18S RNA of the 40S ribosomal subunit and eIF2 [17].” Reference [17] is a crosslinking experiment. If one does not know that -3 or +4 sites are important, then few researchers would attribute site importance to outcomes of crosslinking experiments. In fact, there is hardly any consistent support for the +4G site as important for translation initiation, although the -3R has been substantiated in many experiments. It has been argued before (PMID: 17285142) that the preponderance of +4G site is simply due to the following. First, more than half of the proteins undergo N-terminus methionine excision (NME). Second, NME requires a small and nonpolar amino acid (which typically means glycine or alanine). As glycine and alanine are encoded by GGN and GCN, we have AUG GGN or AUG GCN, i.e., a preponderance of +4G. One also notes that highly expressed genes do not exhibit a preference for +4G.”

Response: We completely agree with the reviewer and we apologize for this oversight. We have now modified our statement. This results in a large modification of the description of the -3 and +4 positions of the Kozak context with additional references.

The sentence: “The positions at − 3 and the + 4 (according to the position of the AUG codon) are determinant because they directly interact with the 18S RNA of the 40S ribosomal subunit and eIF2 [17]has been removed and replaced by the following paragraph:

“It has been reported that the nucleotides upstream the AUG codon, especially the purine at position -3 (according to the position of the AUG codon), are important for efficient initiation [13, 17-19]. UV cross-linking experiments in 48S pre-initiation complex (PIC) strongly suggested an interaction between eIF2α and the -3 purine [20]. This was confirmed later by cryo-EM analyzes of PIC indicating that the arginine residues (Arg55 and Arg57) of eIF2α are located at close proximity of the mRNA nucleotides upstream to the AUG codon, including the -3 position [21]. These data are consistent with the role of eIF2 in AUG codon recognition [22]. However, the influence of the nucleotide in +4 position is rather controversial. Some experiments showed that mutation of the guanosine reduces translation initiation efficiency [13] but it was shown by analysis of all possible combinations using a FACS-seq approach that the +4G was not always the optimum [23]. In addition, an alternative explanation was suggested and it was based on the amino acid constraint on the second codon that participates in post-translational modification such as the N-terminus methionine excision [24, 25].“

Point 5: “Some inferences are weak, e.g., “Clustal multiple sequence alignment of the different 5’UTRs (gRNA and sgRNAs) established from SARS-CoV-2 variants (alpha, beta, delta, gamma and omicron) showed a very high conservation of these sequences (Figure S1) suggesting that RNA cis-acting elements within the 5'UTR are of importance for either replication or translation or both”. This alignment means very little. The sequence variation is comparable to the rest of the SARS-CoV-2 sequences, i.e., the 5’ UTRs are NOT more conserved than other part of the viral genome (They are actually less conserved than the coding sequences) and do not suggest that “RNA cis-acting elements within the 5'UTR are of importance for either replication or translation or both”. By the way, MAFFT (PMID: 23329690) and MUSCLE (PMID: 15318951) are much better than Clustal.”

Response: We understand the point of the reviewer and we prefer to remove the sentence “Clustal multiple sequence alignment of the different 5’UTRs (gRNA and sgRNAs) established from SARS-CoV-2 variants (alpha, beta, delta, gamma and omicron) showed a very high conservation of these sequences (Figure S1) suggesting that RNA cis-acting elements within the 5'UTR are of importance for either replication or translation or both” and the associated supplemental figure from the original manuscript. The sequence alignments of the SARS-CoV-1/2 and MERS-CoV 5’UTRs, and the uORF peptides were now performed with MUSCLE as suggested by the reviewer.

Point 5: The discussion of IRESes is also problematic. While the structures of viral IRESes are well documented to help with ribosome recruitment, Eukaryotic IRESes documented so far are generally devoid of secondary structures and they generally have much weaker IRES activities than those observed in viral IRESes. The lack of structure is interpreted to aid the exposure of the initiation AUG to facilitate translation initiation (PMID: 19125192; 17761883). The statement “Despite differences in length, sequence and structures, all IRESes have the characteristics to directly interact with some components of the 43S initiation complex to bring the ribosome close to the initiation codon [43, 46]” is not substantiated by the two cited references.

Response: The reviewer is right; we have modified our manuscript and described viral and cellular IRESes separately. The text and references have been changed as follows:

“Viral IRESes remain the most studied and despite differences in length, sequence and structures, they usually have the characteristics to directly interact with some components of the 43S initiation complex to bring the ribosome close to the initiation codon [54-63]. On the other hand, eukaryotic IRESes concern ~10% of the mammalian mRNAs and they are generally less structured than viral IRESes [51].

Reviewer 2 Report

The manuscript entitled “Translation of SARS-CoV-2 gRNA is extremely efficient and competitive despite a high degree of secondary structures and the presence of an uORF” by Condé and colleagues reports on the structural features of the 5’-UTR from SARS-CoV-2 and other corona viruses and its efficiencies to initiate translation.

The authors summarize the features of 5’UTRs from all known SARS-CoV-2 gRNAs and sgRNAs including their differences, the secondary structures, the presence of a uORFs and possible IRESes. Using multiple sequence alignments, they show some of these features being conserved between different SARS-CoV-2 strains as well as other corona viruses. Furthermore, they obtained convincing data from in vitro experiments using mainly luciferase assays to show that these viral RNAs are translated efficiently in a cap-dependent mechanism, while there are no signs for an active IRES. In contrast, the uORF seems to play a more prominent role for translation regulation in MERS-CoV compared to SARS-CoV-2. Additionally, a polyadenylation seems to be not necessary in SARS-CoV-2, at least in this specific in vitro setting.

These findings confirm a high translation initiation efficiency of SARS-CoV-2’s 5’UTR region and are important to gain insight into translation mechanisms of corona viruses. The observed differences in the 5’UTRs between SARS-CoV-2 and MERS-CoV which results in different translation efficiency will most likely lead to follow-up studies that investigate this finding in more detail. In addition, this manuscript might also be interesting for scientists in the field of basic research of translation initiation mechanisms in eukaryotic cells. The article is well written, and to a large extent supported by references to the state-of-the-art in corona virus research. I therefore highly recommend considering this article for publication in viruses, after revising several minor points:

1.       Line 22 – 23. To what extend is the mechanism “unusual”? Are there any comparable mechanisms already known? In my opinion, this question is not fully addressed in the discussion. 

2.       Figure 1: It is not intuitive to me, what the different colors besides blue and orange indicates. Also, the alignment of the bars to the right seems counterintuitive.

3.       According to table 1 and lines 115 -116, ORF-7a and ORF-8 lack a variable region. What then is orange part in the corresponding bars of figure 1?

4.       Statistics: Throughout the whole manuscript, the statistics applied are not convincing due to the following reasons:

a.       In none of the figures, the exact number of measured values (n) is given. Its only indicated, that there are at least 3.

b.       with an n of 3, the mean +/- SEM is not meaningful and a t-test should not be used since a test for normal distribution and similar variance is not possible.

c.       Normalizing to a control is leading to a variance within the control of zero, which must be corrected when applying a statistical method.

d.       A correction for multiple testing must be applied.

Considering these points, the statistics should be completely revised, and to describe the statistical methods in the methods section clearly.

5.       Figure 3: To speak of significance in the individual experiments (3b – 3e) is not correct (see comment #4), although observing similar results using different methods (S35-methionine quantification, luciferase activity, RT-qPCR) is convincing.

6.       In line 296 – 299, it’s assumed that N-terminal modification did not significantly impact the enzymatic activity of luciferase and therefore this assay is used as readout for protein production in the remaining of this study.  Was this tested explicitly?

7.       Figure 4c: See comment #4.

8.       Figure 5a and b: See comment #4.

9.       Figure 6d-f: See comment #4.

10.   All observations are obtained with a relatively artificial in vitro approach. Is there any evidence, that the translation of viral RNA is comparable in vivo? For example, in material from Covid-19 patients?

11.   It would be more consistent, if from paragraph 3.2. all investigations would have been performed in all the viruses (SARS-CoV-2, SARS-CoV and MERS-CoV). E.g. is the translation in SARS-CoV and MERS-CoV in this setting also cap-dependent but poly-(A) independent? What happens, if the uORF in SARS-CoV is mutated?

Author Response

Reviewer 2

We thank the reviewer for their insightful comments and have incorporated a number of the suggestions resulting in a much-improved manuscript.

Point 1: Line 22 – 23. To what extend is the mechanism “unusual”? Are there any comparable mechanisms already known? In my opinion, this question is not fully addressed in the discussion. 

Response: The mechanism used by the SARS-CoV-2 5’UTR to initiate translation is dependent of the cap structure but not the poly(A) tail while all viral mRNAs are capped and polyadenylated. To our knowledge, no publication has reported a neutral influence of the poly(A) tail in translation which makes this translation initiation unusual. This is now better discussed:

“In the canonical translation initiation mechanism, the interaction between the poly(A) binding protein (PABP) and eIF4G [122, 123] bring the 5’- and 3’-ends of the mRNA together and forms a mRNA’s circularization called the closed-loop model [124, 125] which promotes translation [89, 126]. Interestingly, in our system which is able to recapitulate the cap/poly(A) synergy [89], the poly(A) tail did not provide any translational advantage to the mRNAs that harbor the SARS-CoV-2 5’UTRORF1ab when compared to globin 5’UTR which uses a strict canonical cap-dependent translation initiation mechanism (Figure 5). This particularity is quite unusual in the eukaryotic kingdom and this raises the question of whether this closed-loop conformation takes place on mRNAs that harbor the SARS-CoV-2 5’UTR. The poly(A) tail requirement should be further investigated, in the future, to determine if the PAPB is necessary to promote initiation and still interacts with eIF4G. To our knowledge, a neutral influence of the poly(A) tail, in translation, has never been reported before. Viral and cellular mRNAs without a poly(A) tail were described but they use alternative and specific cis-acting elements to bring the 5’- and 3’ends of the mRNA together [127-132]. Analyzes of the formation of 48S PIC in RRL through a sucrose gradient indicated that it is almost absent on mRNAs harboring the SARS-CoV-2 5’UTR but not with the EMCV IRES [80] already suggesting that the initiation process on SARS-CoV-2 is not canonical. “

Point 2: Figure 1: It is not intuitive to me, what the different colors besides blue and orange indicates. Also, the alignment of the bars to the right seems counterintuitive.

Response: We completely modified the figure 1 which now gives more information to the reader. The cap and poly(A) structures are indicated. All the ORFs are clearly mentioned. The 5’UTR of each viral mRNA is underlined and composed of the leader sequence in black and a variable region (in orange) specific to viral transcript. This is clearly described in the figure legend.

Point 3: According to table 1 and lines 115 -116, ORF-7a and ORF-8 lack a variable region. What then is orange part in the corresponding bars of figure 1?

Response: We thank the reviewer for this remark. The 5’UTRs of ORF-7a and ORF-8 harbor only the leader sequence. We apologize for this mistake in the figure 1 which has been corrected.

Point 4: Statistics: Throughout the whole manuscript, the statistics applied are not convincing due to the following reasons:

  1. In none of the figures, the exact number of measured values (n) is given. Its only indicated, that there are at least 3.
  2. with an n of 3, the mean +/- SEM is not meaningful and a t-test should not be used since a test for normal distribution and similar variance is not possible.
  3. Normalizing to a control is leading to a variance within the control of zero, which must be corrected when applying a statistical method.
  4. A correction for multiple testing must be applied.

Considering these points, the statistics should be completely revised, and to describe the statistical methods in the methods section clearly.

Response: The reviewer is right and we apologize for these mistakes. We have requested the expertise of Omran Allatif (additional author) from the Bioinformatics and Biostatistics Service (BIBS) at the CIRI. When re-examining our data (see figures below), a batch effect was outlined from graphical display. Indeed, experiment Date has had a major effect which must be accounted for in statistical analysis. Thus, pairing our analysis by Date and using a non-parametric test, due to the small number of our replicates, led us to use Friedman sum rank test, which does not assume that the data originate from a particular distribution. Our null hypothesis is, after removing the experiment Date effect, the distribution location of mRNA expression is the same whatever the treatment condition is. When the null hypothesis was rejected, we proceeded to the Conover's post-hoc test to compare each of our treatments to the Control condition.

These analyses are perfectly in line with our initial message that is: (i) the 5’UTRs of the SARS-CoV-2 mRNAs are able to promote efficient translation initiation, especially the long 5’UTR (ORF1ab); (ii) expression from MERS-CoV 5’UTR is lower than expression mediated by SARS-CoV 5’UTRs.

However, we strongly believe that this kind of representation (boxplot with raw data) is quite complex and not adapted to our reader which uses to histogram illustrations. In addition, expression mediated from the 5’UTR of the Globin is our control and referent from each individual experiment. Indeed, several parameters could affect the values of the raw data but the level of expression induced by the different coronavirus 5’UTRs are always conserved in comparison to the Globin 5’UTR. For these reasons, we prefer to keep the histogram representation with the globin condition like the referent set to 100% (fig. 3 and 4). As requested by the reviewer, the number of independent experiments is clearly indicated in the figure legends as follows: the values of figures 3B, 3C, 5, 6 and 7 represent the mean ± S.D. of experiments performed in triplicate on three independent occasions. The number of repetitions is now indicated for the figures 3D (n=4), 3E (n=4), 4C (n=6) and 4D (n=5). As a result, the values are the mean ± S.E.M.

We prefer to remove all the statistical analysis to not overload the figures and, as the reviewer said, “observing similar results using different methods (S35-methionine quantification, luciferase activity, RT-qPCR) is convincing”.

For information, these are the p-values obtained from the statistical tests in the next two tables:

Point 5: Figure 3: To speak of significance in the individual experiments (3b – 3e) is not correct (see comment #4), although observing similar results using different methods (S35-methionine quantification, luciferase activity, RT-qPCR) is convincing.

Response: The reviewer is right. See response to point 4

Point 6: “In line 296 – 299, it’s assumed that N-terminal modification did not significantly impact the enzymatic activity of luciferase and therefore this assay is used as readout for protein production in the remaining of this study.  Was this tested explicitly?”

Response: This subtle N-terminal modification (only one amino-acid downstream the methionine) may alter the enzymatic activity of the renilla luciferase. To exclude this possibility, the same experiment, performed in triplicate from three individual occasions, has been analyzed by S35-methionine labelling (fig.3B) and by luciferase assay (fig.3C). The expression profile from these two experiments are quite identical which strongly suggests that the luciferase data can reflect the level of expression of the protein. The explanation is more explicit in the result section as follows:

“Thus, this N-terminal modification could have an impact on the activity of the luciferase; the same samples have also been quantified by luciferase assay. As the results obtained by reading of the luciferase assays are consistent with S35-methionine quantification (compare Figures 3B and 3C), we assume that this N-terminal modification did not impact significantly the enzymatic activity of luciferase”

Point 7: Figure 4c: See comment #4.

Point 8: Figure 5a and b: See comment #4.

Point 9: Figure 6d-f: See comment #4.

Response to points 7, 8 and9: see response to point 4

Point 10: “All observations are obtained with a relatively artificial in vitro approach. Is there any evidence, that the translation of viral RNA is comparable in vivo? For example, in material from Covid-19 patients?”

Response: This is a good point from the reviewer. It would be very informative to perform additional experiments in SARS-CoV-2 infected cells in the future but it remains challenging experiments because additional translational control will come into play including the nsp1-mediated translation initiation (This is already discussed in the manuscript). Unfortunately, we do not have access to any covid-19 patients’ material.

Point 11: “It would be more consistent, if from paragraph 3.2. all investigations would have been performed in all the viruses (SARS-CoV-2, SARS-CoV and MERS-CoV). E.g. is the translation in SARS-CoV and MERS-CoV in this setting also cap-dependent but poly-(A) independent? What happens, if the uORF in SARS-CoV is mutated?”

Response: We understand the point of view of the reviewer. This work is to understand how SARS-CoV-2 RNAs are translated and what are the main cis- and trans-acting actors that modulate the translation initiation efficiency. A complete comparison of cap, poly(A) and eIFs requirement between the different and recent human coronavirus strains would be of interest in another and specific study.
